# The Effect of Initial Grain Size on the Nanocrystallization of AZ31 Mg Alloy during Rotary Swaging

**DOI:** 10.3390/ma15227979

**Published:** 2022-11-11

**Authors:** Xin Chen, Silong Li, Yingchun Wan

**Affiliations:** 1School of Minerals Processing and Bioengineering, Central South University, Changsha 410083, China; 2Light Alloy Research Institute, Central South University, Changsha 410083, China

**Keywords:** rotary swaging, AZ31 Mg alloy, initial grain size, nanocrystallization

## Abstract

Nanograins were obtained in the AZ31 Mg alloy bars with different initial grain sizes via cold rotary swaging. Microstructure evolution during deformation was investigated through electron backscatter diffraction analysis and transmission electron microscopy studies. The results indicate that initial grain size had little effect on the mechanism of grain refinement during swaging. The nanocrystallization process of the alloys with different initial grain sizes included extensive twinning followed by the further refinement of the twin lamellae through the formation of massive dislocation arrays. However, as the initial grain size decreased, the formation rate of nanograins increased, resulting in a higher degree of nanocrystallization after the same swaging pass. The mean grain size and yield strength of the sample with the smallest initial grain size were about 91 nm and 489 MPa, respectively. The slower rate and lower degree of nanocrystallization in the alloy with a larger initial grain size were mainly attributed to the less grain boundary areas and higher activity of twinning.

## 1. Introduction

Grain refinement is one of the most efficient methods to improve the mechanical properties of metal materials [1]. It is now well-established that significant grain refinement can be obtained through severe plastic deformation techniques, such as high-ratio differential speed rolling [2], equal channel angular pressing [3], and high-pressure torsion [4]. Generally, changing the strain path [5], reducing deformation temperature [6], and increasing strain rate [7] can bring about a better effect of grain refinement in most metal materials. Moreover, the microstructure in the initial billet can also influence the grain refinement process during deformation, through different factors such as the initial grain size [8], artificially formed precipitates [9], shear bands [10], and numerous twins [11] induced prior to the deformation.

Mg alloys, as the lightest metallic structural materials for application in automobiles and aircraft, can achieve a better strengthening effect than steel and Al alloys through grain refinement due to their higher Hall–Petch slope value [12]. In most Mg alloys, the change in the initial grain size can lead to the transformation of slip and twinning behaviors [13,14,15,16,17,18,19,20]. Koike [13] and Agnew [14] indicated small-grained AZ31 Mg alloy could induce the activation of more non-basal slip dislocations by the plastic compatibility stress associated with grain boundaries. Therefore, the non-basal slip dislocation is enhanced by finer initial grain sizes [15] and is even active in the grain interior [16]. Compared with slip dislocation, twinning is more sensitive to the initial grain size [17]. The activity of twinning generally increases with increasing initial grain size, which is attributed to easier interaction with lattice dislocations that “promote” twin growth in larger grains [17,18]. The number of twins per grain is found to increase with increasing grain size [19], while the twin thickness is reported to be independent of the initial grain size [20].

Recently, we successfully prepared bulk nanocrystalline Mg alloys via cold rotary swaging [21]. However, the effect of initial grain sizes on the mechanism of grain refinement during rotary swaging is still unclear. Additionally, the possibility of nanocrystallization by using the initial bar billets with a much larger grain size (≥500 μm) is unknown. In the present work, AZ31 Mg alloy, one of the currently most widely used commercial wrought Mg alloys, was selected as our model material. The microstructure evolution in samples with different initial grain sizes and their effects on the formation of nanograins (NGs) during rotary swaging were studied in detail.

## 2. Material and Methods

### 2.1. Sample Preparation

The alloy ingots with a diameter of 90 mm were prepared by melting pure Mg, pure Al, pure Zn, and Al-10 wt.% Mn master alloy under a CO_2_ and SF_6_ atmosphere and casting in an upright semi-continuous casting machine. Homogenization was conducted in an air furnace at 693 K for 12 h, followed by air cooling. Then, a cylindrical-shaped sample with a height of 500 mm and a diameter of 18 mm was cut from the ingot for further rotary swaging and this bar was denoted as “HS”. The alloy composition of the homogenized billet was measured by an inductively coupled plasma spectrometer and is listed in Table 1.

Two cylindrical-shaped samples with a height of 250 mm and a diameter of 90 mm were cut from the ingot for extrusion. The extrusion was performed using a backward extrusion method under a ram speed of 10 mm/s. The first sample was processed with an extrusion temperature of 623 K and an extrusion ratio of 25. A cylindrical bar with a diameter of 18 mm was obtained and denoted as “E25S”. The second sample was processed with an extrusion temperature of 723 K and an extrusion ratio of 9. A cylindrical bar with a diameter of 30 mm was obtained. Then, its diameter was reduced to 18 mm using a lathe machine, and this bar was denoted as “E9S”.

The HS, E9S, and E25S samples were further performed via multi-pass rotary swaging under the same process at room temperature. Figure 1 shows the schematic representation of rotary swaging. The accumulated deformation degree after each pass is calculated using Equation (1), where S_0_ is the initial and S_1_ is the cross-section after swaging, and the values are shown in Table 2. Compared with that for the E9S and E25S samples, the utmost accumulated deformation degree for the HS sample was only 0.14. Further increasing deformation would lead to the failure of the HS sample, probably due to more casting defects existing in the homogenized sample.
*φ* = *ln*(*S*_0_/*S*_1_)(1)

### 2.2. Microstructural Characterization and Mechanical Property Evaluation

Vickers microhardness tests were conducted using an HMV-G 21DT (Shimadzu, Tokyo, Japan) tester with a load of 4.9 N and a dwell time of 15 s. The interval of the adjacent measurement point was 0.5 mm. Dog-bone-shaped tensile samples with a gauge length of 15 mm and a diameter of 3 mm were prepared from the center of the alloy bars. The tensile tests were performed using an Instron 3369 machine at an initial strain rate of 1 × 10^−3^ s^−1^ at room temperature, with the tensile directions parallel to the feed direction.

The samples for microstructural observation were all cut from the central region of the alloy bars and the observation plane was on the cross-section. Electron backscatter diffraction (EBSD) measurements were conducted using an Helios Nanolab 600i (FEI, Hillsboro, OR, USA) scanning electron microscope (SEM) equipped with the HKL Channel 5 data acquisition and analysis software. Transmission electron microscopy (TEM) and high-resolution TEM (HRTEM) observations were carried out on a Tecnai G^2^ F20 (FEI, Brno, Czech) microscope with an accelerating voltage of 200 kV using a double-tilt specimen stage. The cut disks were first mechanically ground to less than 100 μm in thickness, then thinned by twin-jet polishing to an electron-transparent thickness in an electrolyte containing 1 vol.% nitric acid, 2 vol.% perchloric acid, and 97 vol.% ethanol. The polishing temperature was approximately −40 °C. The average grain size was measured with the Nano Measurer software 1.2. At least 500 grains were measured for the statistical analysis of the grain size.

## 3. Results

### 3.1. Initial Microstructure

Figure 2 shows the microstructure of the as-homogenized and as-extruded samples. The mean grain sizes of the HS-0 sample, the E9S-0 sample, and the E25S-0 sample were about 548 ± 122 μm, 71 ± 27 μm, and 18 ± 5 μm, respectively.

### 3.2. Microstructure Evolution during Swaging

#### 3.2.1. After One-Pass Swaging

Figure 3 shows the comparison of microstructure between the HS-1 sample, the E9S-1 sample, and the E25S-1 sample. In the HS-1 sample, different types of twins were observed, including the {10$\bar{1}$1}-{10$\bar{1}$2} double twins (~1.5%), {10$\bar{1}$1} contraction twins (~2.9%), and {10$\bar{1}$2} tension twins (~23.7%). These twins refined the initial coarse grains into a thin lamellar structure. In comparison, the double twins and contraction twins were hardly observed in the E9S-1 sample. The main type of twins formed in the E9S-1 sample was the {10$\bar{1}$2} tension twins (~24.1%). In the E25S-1 sample, the number of twins observed was much smaller than that found in HS-1 and E9S-1 samples, as shown in Figure 3h.

Figure 4 shows the typical microstructure in the central region of HS-1, E9S-1, and E25S-1 samples. After one-pass swaging, many lamellar structures with lamellar widths ranging from 100 nm to 1 μm were formed in the samples. After randomly selecting ten lamellae for the selected area electron diffraction (SAED) analysis, most of these lamellae were determined to be {10$\bar{1}$2} tension twins (indicated as “T”) with misorientations of about 86° at the interface, as shown in the inset SAED patterns in Figure 4a,d,f. In the HS-1 sample, two fine twins (T_H2_ and T_H3_) with a width of about 10 nm could be observed within the twin lamella T_H1_, as shown in Figure 4a,b. The misorientation of the boundary between the two fine twins and T_H1_ was about 88°, as shown in Figure 4c, which indicates that T_H2_ and T_H3_ are {10$\bar{1}$2} tension twins too. This means that {10$\bar{1}$2}-{10$\bar{1}$2} double twinning could occur in the HS-1 sample, which was not observed in E9S-1 or E25S-1 samples. In the E9S-1 sample, many twin–twin intersections were formed, such as T_E1_ and T_E2_ shown in Figure 4d. Viewed from a single <10$\bar{1}$2> axis, both T_E1_ and T_E2_ could be well imaged, as shown in Figure 4e, indicating that T_E1_ and T_E2_ might correspond to (10$\bar{1}$2) and ($\bar{1}$012) tension twin variants [22]. Compared with HS-1 and E9S-1 samples, most of the twin morphology observed in the E25S-1 sample was nearly parallel, as shown in Figure 4f.

#### 3.2.2. After Three-Pass Swaging

After three-pass swaging, the three groups of samples exhibited different degrees of nanocrystallization. Figure 5a shows that only a small number of nanoscale subgrains (NSGs) were formed in the local region of the HS-3 sample. The proportion of NSGs obviously increased in the E9S-3 sample, as shown in Figure 5b. In the E25S-3 sample, massive NSGs were formed, as shown in Figure 5c. This indicates that a higher degree of refinement could be obtained at the sample with a smaller initial grain size.

Figure 6a,d show the NSGs (indicated as “NSG1” and “NSG2”, respectively) formed in HS-3 and E9S-1 samples, respectively. The grain sizes of NSG1 and NSG2 were ~80 nm. Both the boundary 1 (B_1_) of NSG1 and the boundary 3 (B_3_) of NSG2 had a misorientation of about 78°, which belongs to an abnormal {10$\bar{1}$2} tension twin boundary [23,24]. The deviation of the angle from the common 86° was considered to be a result of the accommodation of local strain after twinning [23,24]. Moreover, NSG1 and NSG2 both had high-angle grain boundaries, namely B_2_ (~11°) in NSG1 and B_4_ (~41°), B_5_ (~29°), and B_6_ (~17°) in NSG2. This indicates that the NSGs formed in the sample with different initial grain sizes showed similar features.

#### 3.2.3. After Five-Pass Swaging

Figure 7a shows that little difference in contrast was observed between the neighboring grains in the E9S-5 sample. This is probably due to the existence of many low-angle grain boundaries, making it difficult to obtain a convincing statistic of the mean grain size. The unsharp boundaries and relatively discontinuous ring in the inset SAED pattern further indicate that many NSGs in the E9S-5 sample did not completely transform into NGs. In contrast, the microstructure of the E25S-5 sample showed well-defined and clean boundaries with uniform contrast at the grain interiors. The mean grain size was ~91 ± 4 nm. The continuous diffraction ring in the inset SAED pattern indicates that the NGs formed in the E25S-5 sample were randomly oriented. Figure 7 indicates that the sample with a smaller initial grain size showed a higher degree of nanocrystallization after five-pass swaging.

### 3.3. Mechanical Properties

#### 3.3.1. Microhardness Distribution after Different Swaging Passes

Figure 8 shows the microhardness distributions along the radial direction of the swaged alloy bars after different swaging passes. The three groups of samples with different initial grain sizes all showed significant improvement in microhardness after swaging. As the swaging passes increased, a gradient distribution of microhardness along the radial direction of these samples could be observed, which showed a higher hardness value at the center than at the edge. In comparison, the HS sample only showed a slightly gradient distribution of microhardness after three-pass swaging. The E9S sample exhibited an obvious gradient distribution of microhardness after three-pass swaging, while the E25S sample showed a remarkable gradient distribution of microhardness after two-pass swaging. More intensive hardening can be observed at the center of the E25S sample, compared with the E9S and HS samples at the early stage of swaging, as shown in Figure 9.

#### 3.3.2. Tensile Properties

Figure 10 shows the stress–strain curves of the homogenized, extruded, and swaged samples in tension at room temperature. The corresponding mechanical properties are listed in Table 3. The yield strength of the HS sample increased from 155 ± 21 MPa to more than 333 MPa after three-pass swaging. For the extruded samples, the yield strength of E9S and E25S samples increased from 186 ± 8 MPa and 202 ± 3 MPa to 460 ± 19 MPa and 489 ± 9 MPa, respectively, after five-pass swaging. Three groups of samples all showed significant improvement in strength after rotary swaging. It is worth noting that HS-3 and E9S-5 samples fractured before reaching the ultimate tensile strength according to the tensile curves. The former was attributed to the failure to eliminate casting defects through hot extrusion, while the early fracture of the E9S-5 sample was considered to be the result of incomplete refinement after five-pass swaging.

## 4. Discussion

### 4.1. The Same Refinement Mechanisms

Figure 3 and Figure 4 indicate that the refinement of the initial grains of all three groups of samples was mainly due to extensive twinning. Figure 11 shows the further segmentation of the twin lamellae in HS-2 and E25S-2 samples. Massive dislocation arrays were formed within the twin lamellae, as shown in Figure 11a,d, which refined the twin lamellae into many nanoscale subgrains. Boundary 1 (b_1_ in Figure 11b) and boundary 3 (b_3_ in Figure 11e) are the typical dislocation arrays formed in the Mg alloys [25], which have a small orientation of ~5°. These dislocation arrays would transform into subgrain boundaries with the increase in dislocation pile-ups [25], such as boundary 2 (b_2_ in Figure 11c) and boundary 4 (b_4_ in Figure 11f) with the misorientation of ~10° and ~8°, respectively. The further formation of high-angle grain boundaries might have resulted from dislocation pile-ups and interactions in the boundaries and subsequent crystal rotation with their neighboring grains, which implies the occurrence of DRX during rotary swaging, as reported in our previous work [21]. Microstructure evolution indicates that the three groups of samples with different initial grain sizes showed the same grain refinement mechanisms during swaging.

### 4.2. Difference in the Nanocrystallization Process

Figure 8 shows that all three groups of samples had a gradient distribution of microhardness along the radial direction after swaging. In our previous work [21], it was proven that the higher hardness value at the center of the swaged alloy bars is attributed to the formation of NGs, which could not be formed at the edge even after five-pass swaging. A more obvious gradient distribution indicated more remarkable refinement in the central region of the alloy bars. In addition, Figure 5 and Figure 7 show that a higher degree of grain refinement was obtained in the sample with a smaller initial grain size after the same swaging process. This indicates that the formation rate of NGs decreased with the increase in the initial grain size during swaging. This difference was probably attributed to the different capacities for activating non-basal dislocations. On the one hand, as initial grain sizes increased, the activity of non-basal dislocations decreased due to smaller grain boundary areas [13]. On the other hand, the sample with a larger initial grain size showed higher activity of twinning, as shown in Figure 3 and Figure 4. Higher activity of twinning would inevitably decrease the number of activated movable dislocations due to the release of the local stress concentration through twinning, while dislocation arrays are reported to stem from the piling up of non-basal dislocations in Mg alloy [26,27]. Lower activity of non-basal dislocations in the sample could lead to more difficulty in the formation of dislocation arrays, which further delayed the refinement of twin lamellae [21] after the same swaging pass. As a result, the formation rate of NGs slowed down in the sample with a larger initial grain size.

## 5. Conclusions

In summary, three groups of AZ31 Mg alloy bars with remarkably different initial grain sizes were processed via rotary swaging at room temperature. The effect of the initial grain size on the grain refinement process was studied based on the observation of microstructure evolution under the same accumulated deformation degree. The following conclusions were drawn:(1)All three groups of samples showed a significant increase in microhardness and strength after rotary swaging. The improvement in hardness and strength was mainly attributed to the formation of nanoscale grains/subgrains in the central region of the alloy bars;(2)The formation mechanism of nanograins during rotary swaging was independent of the initial grain sizes. The initial coarse grains in the three groups of alloys were all segmented via multiple twinning and followed by the further refinement of twin lamellae through forming massive dislocation arrays;(3)The formation rate of nanograins significantly decreased with the increase in the initial grain size. After three-pass swaging, only local nanocrystallization could be obtained in the sample with a much larger initial grain size (~548 μm). After five-pass swaging, incomplete nanocrystallization was still observed in the sample with a modest initial grain size (~71 μm). In contrast, completely randomly oriented nanograins with well-defined and clean boundaries were obtained in the sample with the finest initial grain size (~18 μm). The lower degree of nanocrystallization in the sample with a larger initial grain size was mainly attributed to the higher activity of twinning at the early stage of swaging.

## Figures and Tables

**Figure 1 materials-15-07979-f001:**
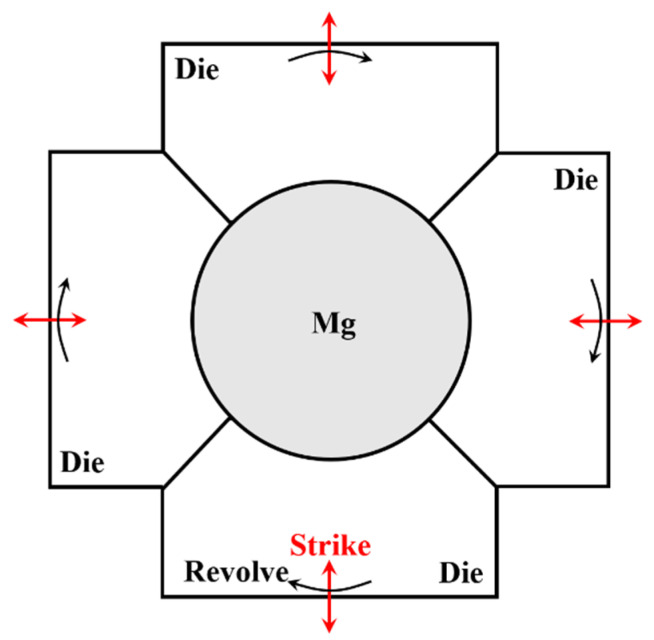
Schematic representation of rotary swaging.

**Figure 2 materials-15-07979-f002:**
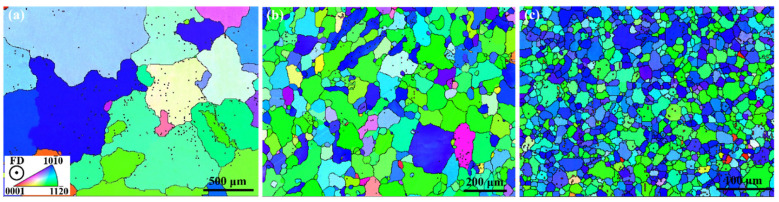
EBSD inverse pole figure (IPF) images of the as-homogenized and as-extruded alloy bars: (**a**) HS-0 sample; (**b**) E9S-0 sample; (**c**) E25S-0 sample. The observation planes are vertical to the feed direction (FD).

**Figure 3 materials-15-07979-f003:**
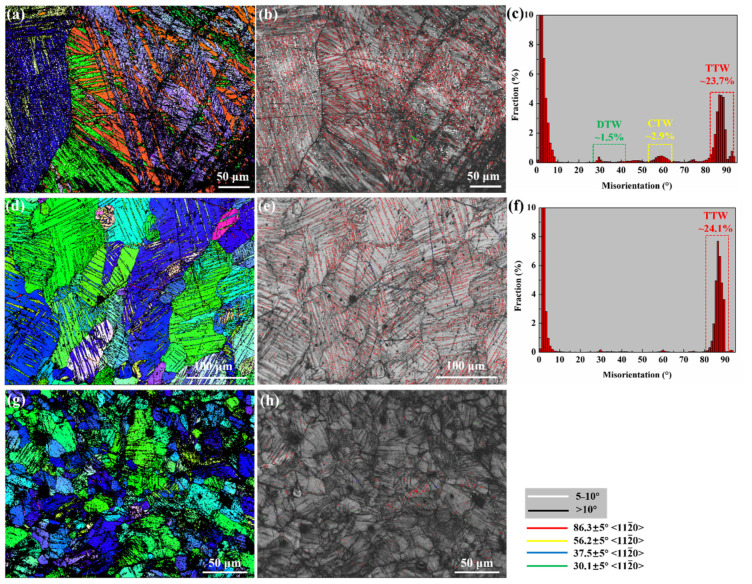
(**a**) EBSD IPF image of the HS-1 sample; (**b**) corresponding band contrast (BC) map of (**a**); (**c**) corresponding misorientation angle distribution map of (**b**). The {10$\bar{1}$1}-{10$\bar{1}$2} double twins, {10$\bar{1}$1} contraction twins, and {10$\bar{1}$2} tension twins are marked as DTW, CTW, and TTW, respectively; (**d**) EBSD IPF image of the E9S-1 sample; (**e**) corresponding BC map of (**d**); (**f**) corresponding misorientation angle distribution map of (**e**); (**g**) EBSD IPF image of the E25S-1 sample; (**h**) corresponding BC map of (**g**).

**Figure 4 materials-15-07979-f004:**
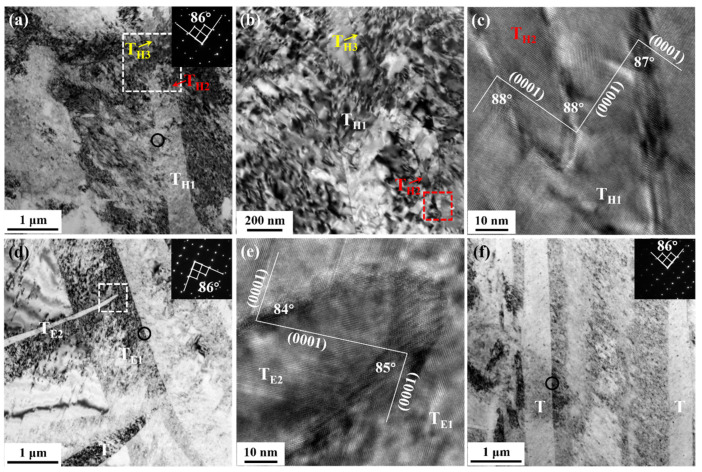
(**a**) Microstructure in the HS-1 sample; (**b**) magnification of the white dotted box in (**a**); (**c**) magnification of the red dotted box in (**b**); (**d**) microstructure in the E9S-1 sample; (**e**) magnification of the white dotted box in (**d**); (**f**) microstructure in the E25S-1 sample. Corresponding SAED patterns of the black circle areas are provided as insets in the micrographs.

**Figure 5 materials-15-07979-f005:**
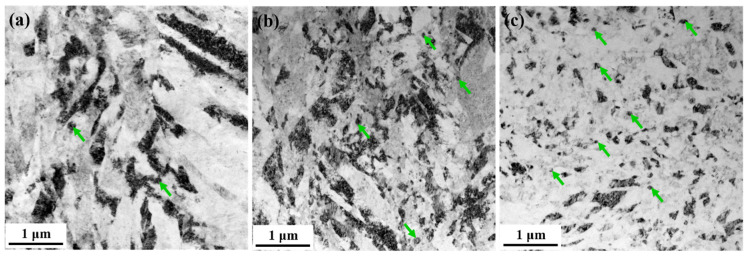
Microstructure in the central region of the alloy bars: (**a**) HS-3 sample; (**b**) E9S-3 sample; (**c**) E25S-3 sample. Nanoscale subgrains are indicated by the green arrows in the micrographs.

**Figure 6 materials-15-07979-f006:**
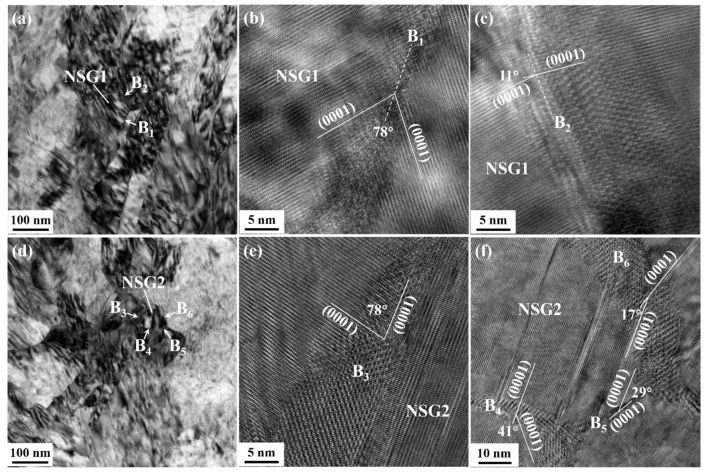
(**a**) A nanoscale subgrain (NSG1) observed in the HS-3 sample; (**b**,**c**) magnified HRTEM images of the (**b**) boundary 1 (B_1_) and (**c**) boundary 2 (B_2_) of the NSG1 in (**a**); (**d**) a nanoscale subgrain (NSG2) observed in the E9S-3 sample; (**e**,**f**) magnified HRTEM images of the (**e**) boundary 3 (B_3_) and (**f**) boundaries 4–6 (B_4_, B_5_, B_6_) of the NSG2 in (**d**).

**Figure 7 materials-15-07979-f007:**
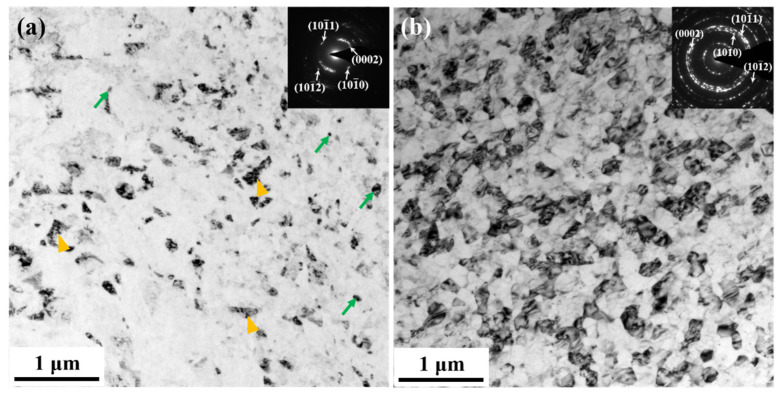
Microstructure in the central region of the alloy bars: (**a**) E9S-5 sample; (**b**) E25S-5 sample. The corresponding SAED patterns are provided as insets in the micrographs. The twin lamellae and nanograins are indicated by the yellow and green arrows, respectively.

**Figure 8 materials-15-07979-f008:**
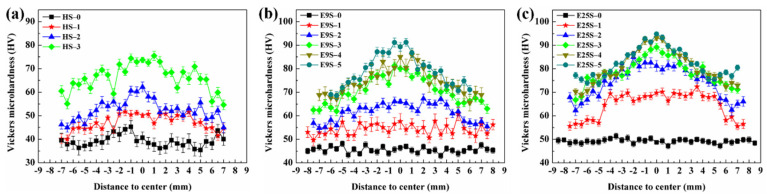
Microhardness distribution along the radial direction after different swaging passes: (**a**) HS sample; (**b**) E9S sample; (**c**) E25S sample.

**Figure 9 materials-15-07979-f009:**
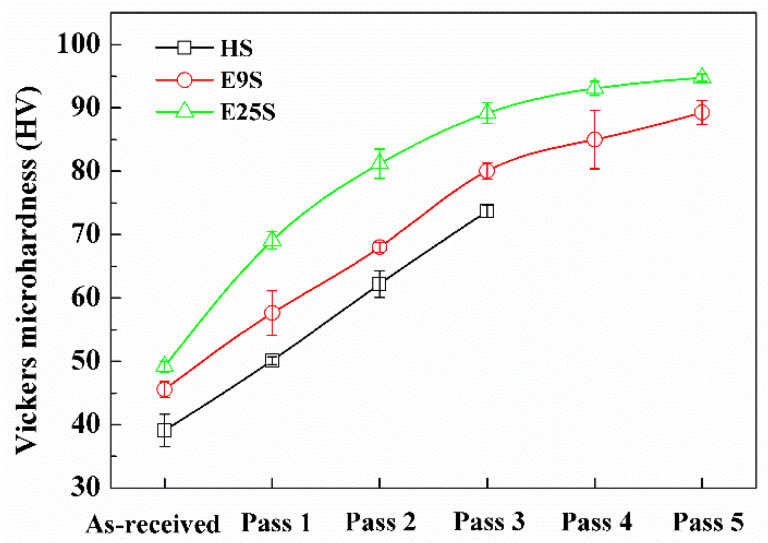
Variation in microhardness at the center of the alloy bars with the increase in the swaging pass.

**Figure 10 materials-15-07979-f010:**
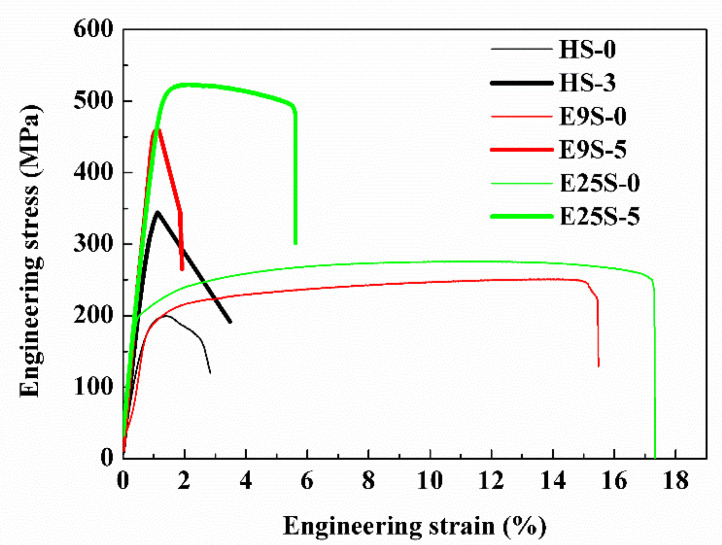
Room temperature tensile stress–strain curves of the as-homogenized, extruded, and swaged samples.

**Figure 11 materials-15-07979-f011:**
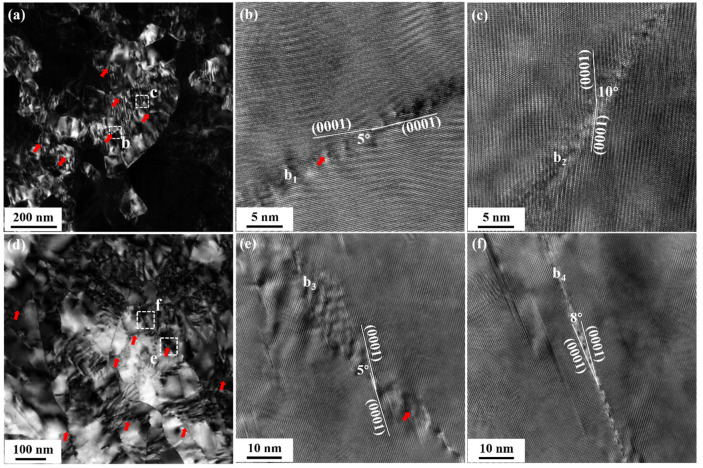
(**a**) TEM dark-field image of the HS-2 sample, showing the segmentation of the twin lamellae by the formation of massive dislocation arrays (red thick arrows); (**b**,**c**) magnified HRTEM images of the white dotted boxes “b” and “c” in (**a**), respectively; (**d**) TEM bright-field image of the E25S-2 sample, showing massive dislocation arrays also formed within the twin lamellae; (**e**,**f**) magnified HRTEM images of the white dotted boxes “e” and “f” in (**d**), respectively.

**Table 1 materials-15-07979-t001:** Chemical composition of the homogenized AZ31 Mg alloy (wt.%).

Element	Al	Zn	Mn	Mg
Composition	3.1	0.9	0.3	Bal.

**Table 2 materials-15-07979-t002:** The same accumulated deformation degree of homogenized sample (HS) and extruded samples with extrusion ratios of 9 (E9S) and 25 (E25S), respectively.

Sample			Swaging Pass	Accumulated Deformation Degree
HS	E9S	E25S		
HS-0	E9S-0	E25S-0	0	0
HS-1	E9S-1	E25S-1	1	0.03
HS-2	E9S-2	E25S-2	2	0.08
HS-3	E9S-3	E25S-3	3	0.14
Failure	E9S-4	E25S-4	4	0.19
Failure	E9S-5	E25S-5	5	0.24

**Table 3 materials-15-07979-t003:** Lists of tensile yield strength (YS), ultimate tensile strength (UTS), and elongation to failure (*ε_ef_*) of the as-homogenized, extruded, and swaged samples.

Sample	YS, MPa	UTS, MPa	*ε_ef_*, %
HS-0	155 ± 21	199 ± 18	2.8 ± 0.9
HS-3	>333	>333	—
E9S-0	186 ± 8	251 ± 6	15.5 ± 0.7
E9S-5	460 ± 19	≥461	≥1.2
E25S-0	202 ± 3	275 ± 2	17.3 ± 0.3
E25S-5	489 ± 9	523 ± 7	5.6 ± 0.8

## Data Availability

The raw data required to reproduce these findings are available from the corresponding author upon reasonable request.

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
