# Peer review of "The Effect of Initial Grain Size on the Nanocrystallization of AZ31 Mg Alloy during Rotary Swaging"

_materials, 2022, doi:10.3390/ma15227979_

Round 1

Reviewer 1 Report

The manuscript presents an interesting study about the microstructure evolution of AZ31 Mg alloy with different initial grain size and their effects on the formation of nanograins. However, the paper needs minor revisions before it is processed further, some comments follow:

Abstract

The abstract must be improved. The abstract must contain information about:

-        Background: Please highlight the novelty of the study;

-        Methods: Describe briefly the main methods used to obtain and characterize the material.

-        Results and conclusions: Indicate the main conclusions or interpretations, also add some quantitative results.

Materials and methods

Introduce the chemical composition of AZ31 into a table. Also, please write how was determined.

Conclusion

Add some suggestion and limitations.

Reviewer 2 Report

The work is very interesting. The research program is properly prepared. The text includes comments and a suggestion (attachment). At work, it is worth paying more attention to the processing method itself, along with a brief description of the technique and its scheme.

Reviewer 3 Report

Notes on the article of Xin Chen, Silong Li and Yingchun Wan “The effect of initial grain size on the nanocrystallization of AZ31 Mg alloy during rotary swaging”

The paper reports results of studying of the microstructure and mechanical properties of AZ31 after different regimes of cold rotary swaging. The authors showed that an increase in the values of mechanical characteristics was happened in all types of processing. It was also shown that the mechanism of microstructure formation after rotary swaging did not depend on the initial grain size. The advantage of the work is a detailed study of the microstructure of the alloy by various methods. It is an interesting and well-written report, which should be published after revisions that are listed below:

1) The scheme of the rotary swaging process used in the work should be added in the "Material and methods" section. The temperature of cold rotary swaging should be also indicated.

2) During the research, the authors compare the studied values with the initial as-cast state. However, it was written in "Material and methods" section: Homogenization was conducted in an air furnace at 693 K for 12 h, following by air colling". Why did the authors not compare with homogenized state? It should also write "by air cooling" instead of "by air colling"

3) How many samples were examined to study the mechanical properties?

4) All calculated values (for example, grain size) should be presented as m ± M, where m is the average value, M is the measurement error.

5) Which area of the sample was studied for analysis of the microstructure of the alloy after swaging? It is known that the structure of the sample after rotary swaging (especially at low degrees of deformation) is not uniform along the sample diameter. In which direction of the rod was the microstructure investigated?

6) Did the authors observe the precipitation of Mg17Al12 particles? It is indicated that the alloy was pre-homogenized and then extruded at 350 and 450 °C. How was the alloy cooled after deformation? Is it possible that the particles precipitated during cooling or deformation? The presence of these particles should affect structure formation

7)    The measurement error should be added to Figure 7.

Round 2

Reviewer 3 Report

The authors carefully revised the article and answered all questions. I recommend to accept the revised version of the article after minor revisions:

The mechanical properties values (Table 3) should be also presented as m ± M, where m is the average value, M is the measurement error.

Author Response

The mechanical properties values have been modified to m ± M in Table 3, as shown in the revised manuscript.